# The Interaction between African Easterly Waves and Different Types of Deep Convection and Its Influence on Atlantic Tropical Cyclones

**Bantwale D. Enyew** [1] **and Ademe Mekonnen** [2,*]

1    Applied Science and Technology PhD Program, North Carolina A&T State University, Greensboro, NC 27411, USA; bdenyew@aggies.ncat.edu
2    Department of Physics, North Carolina A&T State University, Greensboro, NC 27411, USA
*    Correspondence: amekonne@ncat.edu; Tel.: +1-336-285-2222

**Abstract:** This study revisited the association of African easterly waves (AEWs) to Atlantic tropical cyclone (TC) development using weather states (WSs) from the International Satellite Cloud Climatology Project, National Hurricane Center best track hurricane data (HURDAT2), and reanalysis products. The WS data are used as a proxy for two different types of deep convection. This study covers July–October 1984–2009. Statistical analysis based on HURDAT2 and objectively tracked AEWs has shown that a small fraction (~20%) of the AEWs that propagate from Africa serve as TC precursors. About 80% of the AEWs from the continent were non-developing. As in the past work, our study showed an important difference between developing and non-developing AEWs. Composites based on developing AEWs revealed well-organized large scale deep convection (one type, composed of mesoscale systems and thick anvil clouds) is tightly coupled to the AEW trough, while scattered, less well-organized deep convection (second type, isolated cumulonimbus and cumulus congestus clouds) dominated a large area downstream of the developing AEW trough. Developing AEWs propagate westwards while strengthening. In contrast, non-developing AEWs showed that the peak well-organized deep convection is located either behind (to the east of) or far ahead (to the west) of the AEW trough (peaks values are not in close proximity). Moreover, well-organized deep convections associated with non-developing AEWs were weaker than those associated with developing AEWs. The results indicated that convective activity ahead of the non-developing AEWs is weak. Positive relative humidity (RH) anomalies dominate the area around AEWs and downstream over the main TC development region. In contrast, negative RH dominated the main TC development region ahead of non-developing AEWs, suggesting an unfavorable environment downstream of the AEWs. The results also showed that developing AEWs maintained stronger features in the lower and middle troposphere, while non-developing AEWs exhibited weaker structures, in agreement with past work.

**Keywords:** African easterly waves; tropical cyclones; weather states

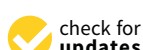

## 1. Introduction

African easterly waves (AEWs) are key boreal summer weather systems that are associated with daily rainfall and deep convection over Africa, e.g., [1–4]. Most of the AEWs originate over East Africa in association with deep convection, e.g., [5,6], and strengthen as they propagate over West Africa in association with the mid-tropospheric easterly jet stream, e.g., [7]. AEWs are characterized by a phase speed of 8 m/s, a period of 3–5 days, and a wavelength of about 2500–4000 km [7–9].

It is well documented that AEWs serve as precursors for the generation of Atlantic tropical cyclones (TC), e.g., [10,11]. Although slightly different numbers are reported in the literature, about half of all Atlantic TCs, e.g., [12], and over 85% of intense hurricanes develop in association with AEW activity, e.g., [13,14]. Recently, Russell et al. [15] noted

that 61% of TCs in the period of 1995–2015 originated directly from AEWs. In general, a significant portion of Atlantic TCs is related to AEWs.

About 50–60 waves per year develop over North Africa, and most of these propagate into the Atlantic, e.g., [12,16]. However, a small fraction of the AEWs that crossed the West African coast served as precursors to TCs. For example, Avila et al. [16] found that about 18% of AEWs are associated with TC genesis. Similarly, Agudilo et al. [17] reported that about 14% of AEWs develop into TCs. It remains to be established as to why only a fraction of the AEWs do develop into tropical cyclones, while most do not. Despite several decades of research, much remains to be known about the relationship between AEWs and TC genesis, e.g., [8,15,18–20].

The role of AEWs on the genesis of Atlantic TCs depends on the large-scale environment that waves propagate through (warm sea surface temperature, weak vertical wind shear, moisture in the middle and upper troposphere, etc.; e.g., [21]). The characteristics and structure of the waves themselves are also found to be important factors, e.g., [17,19,22]. Additionally, Hopsch et al. [19] identified the intensity and amplitude of AEWs as important factors to TC genesis in the eastern Atlantic main development region. Agudelo et al. [17] suggested the amplitude of AEWs entering the Atlantic Ocean is a major determinant for TC genesis and that the TC genesis potential increases when the wave is preceded by pre-existing large-scale convergence and moist convection.

Brammer and Thorncroft [23] reported that AEWs associated with higher moisture content are more likely to develop storms downstream as compared with waves with lower moisture content. Several others, e.g., [24,25], highlighted the important role of moisture associated with propagating waves and pre-existing disturbances ahead of an AEW that is exiting or about to exit coastal West Africa. Hopsch et al. [22] further suggested that developing AEWs are associated with a distinctive cold-core structure over West Africa, increased convective activities over the Guinea highlands, and enhanced low-level vorticity (see also [6]). Such waves are likely to transform into a more warm-core structure in the vicinity of the eastern Atlantic and are likely to spawn TCs. Non-developing AEWs are associated with dry signals ahead of the trough from the middle to upper levels of the troposphere, e.g., [22]. Other studies, e.g., [26–29], using satellite data, suggested that developing waves are distinguished by an association with expansive rainy cloud pixels and a large area coverage of cold cloud tops. The cold cloud top and area covered by cold clouds in the above studies were identified using the satellite observed equivalent brightness temperature ($T_B$) [28]. For example, Leppert et al. [28] suggested that waves associated with moist convection and a larger area of convection (identified by TB $\leq$ 240 K) are critical for the development of a tropical storm. Additionally, Nunez-Ocasio et al. [29], based on a tracking algorithm developed in Nunez-Ocasio et al. [30], quantified the area coverage of mesoscale convective systems (MCSs) and suggested that waves coupled with a large area MCS are important for TC genesis.

Leppert et al. [28] noted that in the days leading up to tropical storm genesis, the fractional coverage of convection associated with the wave trough and with northerlies increases, while the intensity of convection (as measured by lightning flash rates) decreases. They concluded that area coverage is more relevant to TC genesis than convective intensity (see also [29]). However, as suggested in recent studies, $T_B$ and OLR data are not explicit measures of deep convection. Both $T_B$ and OLR cannot isolate cold cloud tops associated with suppressed convective clouds such as upper tropospheric cirrus, e.g., [31,32]. In this study, we use weather state (WS) data from the International Satellite Cloud Climatology Project (ISCCP) to investigate the wave–convection interaction. The WS data provide a unique opportunity to discriminate deep convective states from non-convective cold clouds. The WS approach provides a prospect to identify different types of deep convection, i.e., deep convection associated with mesoscale systems vs. deep convection associated with scattered and isolated convection. This approach allows us to investigate convective organization and development through a "transition" between different types of deep convection, e.g., [32,33]. WSs also provide important information about the area coverage

and vertical depth of convective systems, e.g., [34]. The findings of Leppert et al. [28], for example, can be more refined by using a WS dataset.

Dieng et al. [24] studied AEWs associated with TC genesis between coastal West Africa and the eastern Atlantic near 40° W. They noted that about 90% of tropical storms that formed near coastal West Africa were associated with AEWs coupled with mesoscale deep convection. Dieng et al. [24] further noted that most of the AEWs that are linked to tropical storm genesis are preceded by a trough some 2000 km farther west from a trough axis at the West African coast. Here, we expand the study area to include the western Atlantic, identify all AEWs that cross the coast, and study the fraction of AEWs that turn into a TCs. The interaction between AEWs and mesoscale deep convection will be studied using ISCCP data. We investigate how the coupling between AEW phases and different types of deep convection evolve over time. We also investigate how different types of deep convection and wave phases interact, separately, for developing and non-developing wave types.

To achieve our objective, therefore, we use the ISCCP and European Centre for Medium-Range Weather Forecasts (ECMWF) datasets to produce long-term climatology (July–October 1984–2009) of AEW characteristics, structure, and its association with convective activities for developing and non-developing AEWs. The official Atlantic hurricane season is from June–November, but the climatological peak activity is from August to October. Thus, this study considers the July–October season.

## 2. Data and Methodology

### 2.1. Data Sources

The major datasets used in this work are drawn from the National Hurricane (NHC) "best track" data archive, from the ECMWF and from the ISCCP Version D archive.

### 2.1.1. National Hurricane Center Best Track Data

The National Hurricane Center (NHC) "best track" hurricane database (HURDAT2) is used to extract information on TC genesis and TCs' further development. TCs in this study include tropical depressions, storms, and hurricanes. The HURDAT2 database, spanning from 1851 to present, includes information on storm tracking and intensity and reports a summary for each named tropical storm [35]. Information on developing waves and the dates that they crossed the West Africa coast were obtained from the HURDAT2 archive (available: https://www.nhc.noaa.gov/data/?#hurdat, accessed on 1 July 2021). These data have information on African easterly waves (AEWs) that are associated with TC genesis. We refer to these as developing. Those AEWs identified using a tracking technique but not linked to TC genesis are referred to as non-developing, e.g., [22].

### 2.1.2. ERA Interim

The large-scale environment, including dynamic and thermodynamic variables, is derived from the ECMWF Interim analysis (ERA Interim) dataset. ERA Interim is available at different horizontal resolutions. Easterly wave detection and tracking (Section 2.2.1) is performed on a higher resolution 0.75° × 0.75° relative vorticity dataset. We also use the 1.5° × 1.5° global grid [36] for investigating the climatology of large-scale circulation and humidity environments. The study period covers 26 years of data from 1984 to 2009, matching the ISCCP products.

### 2.1.3. ISCCP Weather State (WS) and Infrared Weather State (IR-WS) Data

The WS data from ISCCP archive are used to characterize different types of deep convection. A growing body of published work has shown the advantage of using WS data in comparison with cold cloud brightness temperature ($T_B$) and/or outgoing long-wave radiation (OLR) to study the interaction between atmospheric waves and different types of deep convection, e.g., [32,33]. Briefly, WSs are joint frequency distributions of cloud top pressure (CTP) and cloud optical thickness ($\tau$) identified based on mesoscale features in the horizontal distributions of cloud properties for various regions

of the world. We note that the CTP-τ distributions (histograms; Figure 1) are based on nine basic cloud classifications shown in the Supplementary Material (additional information in online Supplemental Material Figure S1). The ISCCP cloud classification describes where each cloud forms in the atmosphere (low, middle, and high-level clouds; Figure S1). The nine-cloud classification is similar to the classic World Meteorological Organization's cloud classification convention (see https://public.wmo.int/en/media/news/international-cloud-atlas, accessed on 13 December 2021).

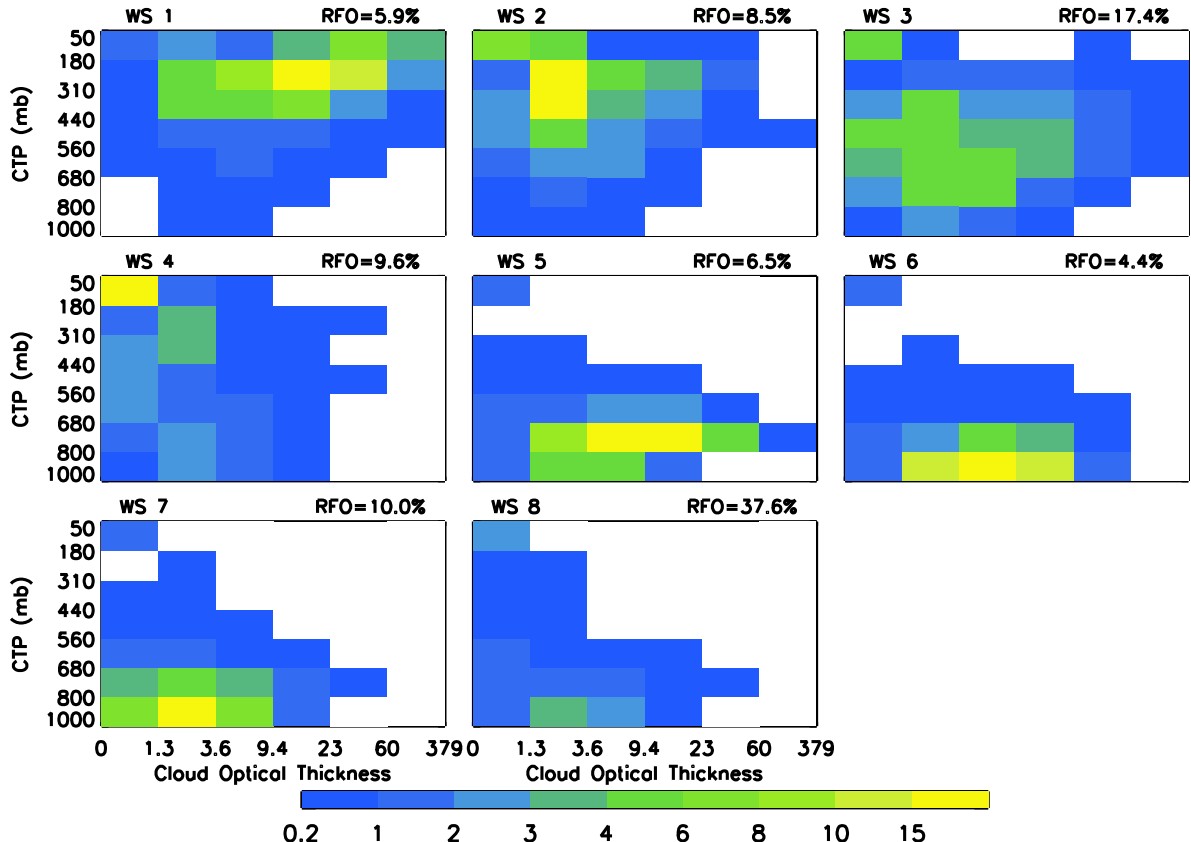

**Figure 1.** The frequency of occurrence of cloud-top pressure-optical thickness patterns (WSs). The histogram is averaged for global subtropics (±35° latitude) and for the 1983–2004 time period derived from ISCCP D1 data. The WSs 1 to 8 are listed from left to right and top to bottom, where most convective states are in the top row and less convective states are in the middle and bottom rows. The WS relative frequency of occurrence (RFO) over the whole domain is given at the top right corner of each plot. From Mekonnen and Rossow [31].

An important aspect of WSs is they are associated with distinct atmospheric conditions [33,37,38]. Based on CTP and τ, past works, e.g., [34,39], have identified different WS categories, different types of deep convection and suppressed convective states, for the different regions of the globe. For example, Mekonnen and Rossow [31,32] used WS categories for the extended global tropics (±35 Latitude) to study the interaction between African easterly waves and deep convection. The CTP-τ pattern analysis for the extended global tropics produces eight distinct WS categories (identified as WS1–WS8; Figure 1). The fractional coverage of mesoscale domains (indicated by colors in Figure 1) denotes the area coverage by each pair of categories (or cloud regimes) with the CTP-τ properties/higher cover fraction (tending towards yellow) covering a larger area of the 2.5° × 2.5° grid cell. Thicker (τ > 23) and higher-topped (CTP < 440 hPa) clouds are associated with deep convection (on the right side of the histograms). Moderately thick and higher-topped clouds are associated with mesoscale anvil clouds. Higher-topped but optically thin clouds on the upper left side of the histograms are related to upper-level cirrus (left in the middle panel).

Three of the WS categories describe three different types of deep convection (represented by WS1, WS2, and WS3; top panel) and the remaining five characterize suppressed convection (WS4–WS8), e.g., [31]. WS1 categories represent the highest frequencies of high-topped clouds (in the upper troposphere above 300 hPa) and by their large optical thickness ($\tau > 23$). These clouds are associated with large-scale, well-organized convection associated with mesoscale convective systems (MSCs). WS2 categories represent high-topped and relatively lower incidences of large $\tau$ and are associated with thick anvil clouds. WS3 categories identify lower incidences of $\tau$, those cloud regimes in the middle troposphere around 400 hPa, and isolated deep convection. WS3 categories characterize scattered, lower scale deep convection (including isolated Cb clouds). WS4 categories represent thin cirrus in the upper troposphere. WS5–WS8 represent various marine and boundary layer clouds. WSs are available at 2.5° grid and at 3-h intervals from June 1983–2009 for the extended global tropics.

WS data are available only for the daytime, as $\tau$ is unavailable during nighttime. However, the ISCCP provides CTP histograms for each grid at all times of the day, where IR retrievals are used. Tan et al. [40] and Tan Jakob [41] developed a three-step process to produce WSs using IR information, referred to as infrared WS (IR-WS). These are: (1) the original WS data are reproduced for each 3-h time step, (2) for each original regime, the corresponding IR-only CTP histograms are determined, and (3) every CTP-histogram is assigned to its new regime by the shortest distance. Based on these, three distinct regimes that are proxies for active convection and 4 regimes that characterize suppressed convection were identified. The three active convective states are referred to as IR-WS1, IR-WS2, and IR-WS3. The IR-WS1 regimes are identified by highest incidences of large $\tau$ and high-topped clouds dominated by deep convective cloud clusters with deep stratiform layers. IR-WS1 regimes correspond to WS1. The IR-WS2 regimes are a lower area coverage of thick clouds with high incidences of thick cirrus anvils. IR-WS2 regimes correspond to WS2. We note that the IR-WS2 clouds exist together with IR-WS1, and we combine them in our analysis (IR-WS12) [32,42].

IR-WS3 regimes are clouds of lower incidences of large $\tau$ and high-top clouds having characteristics of deep convection. IR-WS3, corresponding to WS3, characterize isolated deep convection. Thin cirrus (IR-WS4 type) is also identified in the algorithm but will not be used here. Tan et al.'s [40] classifications also include trade cumulus and various stratocumulus regimes, collectively referred to as suppressed regimes (IR-WS5-IR-WS8). Details of the IR-WS dataset can be found in Tan and Jakob [40] and Tan et al. [41]. The IR-WS has been successfully used to study African easterly waves and convection studies, e.g., [31,32,43,44], and the Madden–Julian oscillation and convection [45]. Here, we use IR-WS12 and IR-WS3 to characterize two different types of deep convection for the 1984–2009 period.

To provide context, the climatology of deep convection, 700 hPa relative vorticity, 700 hPa relative humidity (RH), and 700 hPa wind flow are presented for July–October (JASO) 1984–2012 (Figure 2). The IR-WS12 type of deep convection that is associated with MCS activity is seen over the highlands of eastern Africa, Cameroon–Nigeria, and western coastal regions of Africa (Figure 2a). The maximum positive relative vorticity is in the same zone as the peak of well-organized and large-scale deep convection. Maximum IR-WS12 coincides with peak rainfall climatology. Peak IR-WS3 types of deep convection (scattered and small-scale convective activity) are located either to the south or to the west of maximum IR-WS12 (Figure 2b). The mean relative vorticity is at maximum over central and eastern Africa, the initiation region of the AEWs [5,6,32], and off the coast of West Africa [9,46]. The axis of maximum relative vorticity between 10°–12.5° N is indicative of the AEW track, e.g., [47]. The maximum relative vorticity areas and peak IR-WS12 are co-located (Figure 2a). As will be discussed in more detail later (Section 3), the availability of middle tropospheric moist environment ahead of a wave trough is an indicator of a developing AEW, e.g., [22]. As shown in Figure 2c, 700 hPa high RH values (>70%) are coincident with peak IR-WS12, the deepest and large-scale convection, and located over

high topographic areas and over coastal West Africa. Strong winds are also seen near the climatological position of the middle tropospheric African easterly jet.

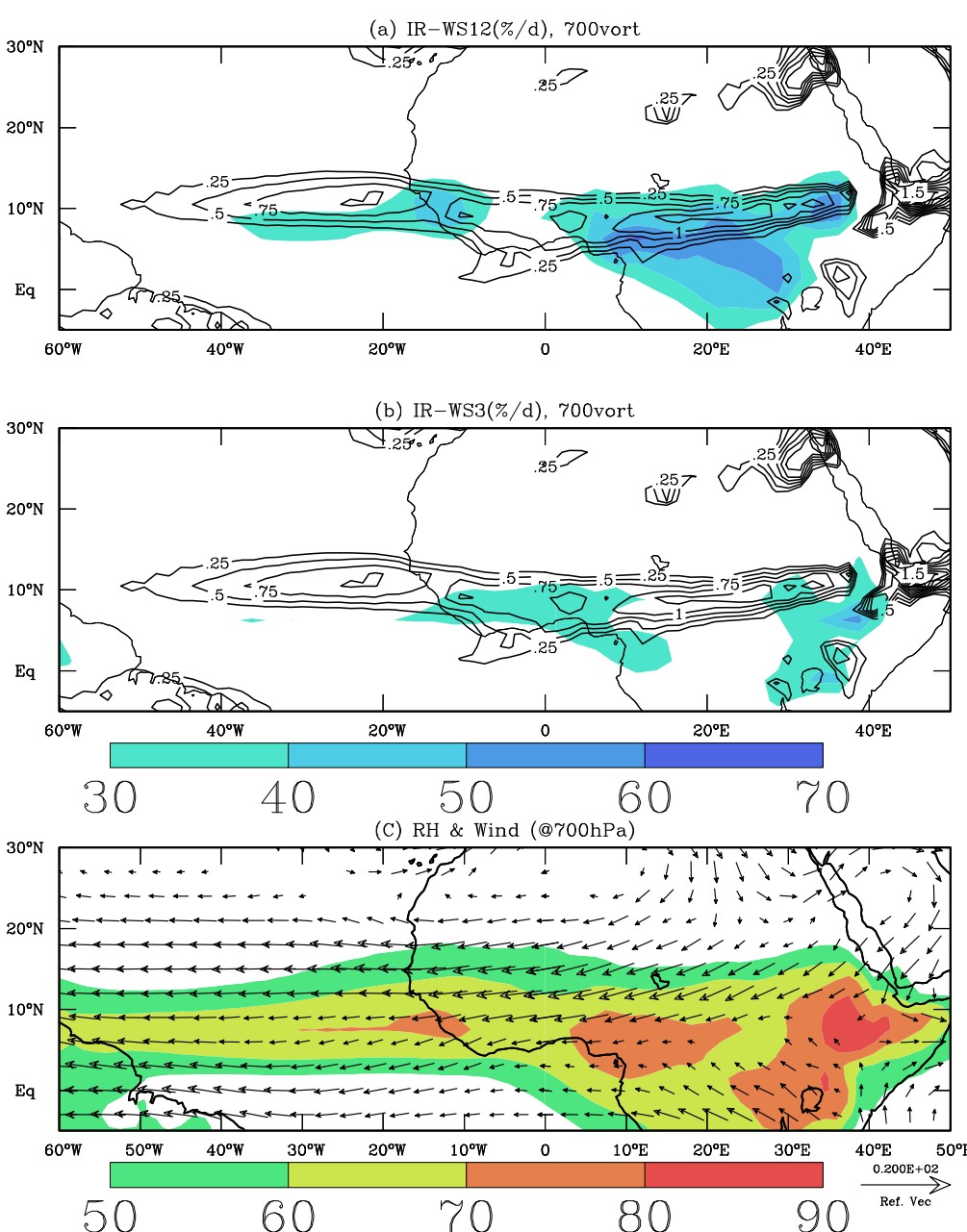

**Figure 2.** The July–October (JASO) 1984–2012 averages: (**a**) IR-WS12 (large scale well-organized deep convection) and 700 hPa relative vorticity; (**b**) IR-WS3 (scattered less well-organized deep convection) and 700 hPa relative vorticity; (**c**) 700 hPa relative humidity and horizontal wind. IR-WS12 and IR-WS3 are frequencies of occurrences per day (shaded as shown below the middle panel), and relative vorticity is contoured every $0.25 \times 10^{-5} \, \text{s}^{-1}$. Mean relative humidity in panel (**c**) is shaded (%). Wind vectors are plotted every 3° and weaker winds (<10% of the maximum wind) are suppressed for clarity. The magnitude of the reference vector is 20 ms$^{-1}$.

### 2.2. Analysis Methods

2.2.1. Detecting and Tracking of AEWs

In the literature, various methods of identifying and tracking AEWs have been employed. Some studies manually tracked and identified AEWs using Hovmöller diagrams, e.g., [26,48]. However, this method can be subjective, time consuming, and difficult to

apply for extended periods of time. Past work, including Thorncroft and Hodges [47] and Hopsch et al. [19,22], has used automatic tracking methods to study AEW genesis and lysis, including variability. Automatic tracking is less tedious, can limit subjectivity, and can be used for an extended period of time. Several different objective and automatic tracking methods have been proposed in the literature. Belanger et al. [49] have constructed an easterly wave database using different reanalysis products for tropical Africa, the Atlantic, and the East Pacific. More recently, Nunez-Ocasio et al. [29] developed an automatic objective tracking algorithm to track and investigate the AEW–moist convection relationship (see also [23]). In a follow-up paper, Nunez-Ocasio, et al. [30] provided a comprehensive analysis of developing and non-developing AEWs and their connection with different types of moist convection. Our objective in this work is to investigate the dynamic property of AEWs using relative vorticity and different types of deep convection using ISCCP weather state products. We will not use the Belanger et al. [49] dataset, as the scope of this study is limited to waves that cross coastal West Africa. The Nunez et al. [30] product is not used for the same reason.

Thorncroft and Hodges [47] used an automatic tracking of vorticity centers in the middle and lower troposphere to study AEW variability and their interaction with Atlantic TCs for the summer of 1979–1998. The method allowed them to identify maximum areas of wave activity south of 15° N near 600 hPa (the region of summertime deep convection) and poleward of 15° N near 850 hPa over West Africa. The automatic tracking method in conjunction with genesis information from the National Hurricane Center archive provides an opportunity to study the role of AEW activity on Atlantic TC genesis [22,47].

We use an objective tracking method following Hodges (50,51) and Thorncroft and Hodges [45]. The automatic tracking method first introduced by Hodges [50] and further improved by Hodges [51] has been used to study AEWs and TC genesis in several studies, including Hopsch et al. [19,22]. In this study, AEW tracking has been performed within the Atlantic peak tropical cyclone activity period from July to October using ERA Interim products. The data were first averaged at a 6-hourly timestep over three pressure levels (850 hPa, 700 hPa, and 600 hPa), where easterly wave activities are strong (e.g., [2,8,18,28]). Past work, e.g., [19,47], applied various criteria for strength (relative vorticity of at least $+0.5 \times 10^{-5}$ s$^{-1}$), for longevity (minimum lifetime of 2 days), and a minimum east-west propagation distance of 10° to be considered as a valid track. The method detects systems that have closed vorticity contours and excludes weaker waves that do not support coherent high vorticity centers [47]. However, a brief observation of AEWs that are linked with TC genesis in the HURDAT2 shows that, in many instances, weaker AEWs (peak relative vorticity $< 0.5 \times 10^{-5}$ s$^{-1}$) can spawn TCs. To account for such possibilities, the relative vorticity threshold is lowered to $+0.2 \times 10^{-5}$ s$^{-1}$. This is designed to avoid the undercounting of developing waves. Other thresholds (propagating for >2 days and zonal scale longer than 10°) are as in Thorncroft and Hodges [47] and Hopsch et al. [19]. In sum, AEW tracking and identification are performed on a mean daily relative vorticity using the Hodges [50,51] automatic tracking method. In the process, those waves that crossed the West African coast between 20°–15° W were retained. This study area is similar to that of Hopsch et al. [22]. The study area is west of the Guinea highlands (the Fouta Djallon mountain chain). As suggested in past work (e.g., [6,15,22]), the interaction with the mountain range could provide an additional source of moist deep convection that could enhance AEW growth and sustenance. Note that the method retains only mature waves that are fully transitioned to the coastal region of West Africa. We discriminate waves that develop into TCs and those that fail to develop using the tracking method and the information from HURDAT2. When the dates of the waves crossing the West African coast match with the dates from the HURDAT2 archive for named tropical cyclones, the waves are considered as developing, otherwise the waves are non-developing.

2.2.2. Composite Analysis

Composite maps are prepared using the information in the HURDAT2 archive and the automatic tracking method adopted for this study. Two timeseries, separately, for developing and non-developing AEW cases, were constructed to form composites. As discussed above, dates of named storms associated with AEWs were identified and cross-checked with automatic tracking to separate developing waves from non-developing waves. For developing waves, day 0 is the time at which a tracked AEW that crossed the West African west coast matches the TC genesis information in HURDAT2. Day 0 for non-developing refers to those days that AEWs were tracked and identified using the tracking method but did not have corresponding information that would associate them with named storms in the HURDAT2 archive. Composites of relative humidity, relative vorticity, wind shear, and horizontal wind from ERA Interim, including infrared weather states (IR-WSs), were examined.

As in past work, e.g., [9,22,28], composites are used to examine the mean structure and evolution of AEWs, separately, for developing and non-developing AEWs. However, in this paper, since the differences between developing and non-developing waves, including horizontal and vertical structure and environmental states associated with AEWs, have been extensively reported, e.g., [15,22], we will present selected background analysis to provide context and emphasis on the results based on IR-WSs.

## 3. Results and Discussion

### 3.1. Tropical Cyclone and AEW Statistics Summary

Based on the Hodges automatic tracking scheme and our criteria outlined above, the total number of AEWs that crossed the West African west coast during July–October of 1984–2009 was 991 (Table 1) c.f. [22]. Of these, 191 AEWs served as precursors to TC development (Table 1) and hence are referred to as developing AEWs, while the remaining 800 were considered as non-developing.

**Table 1.** Total numbers of tropical cyclogenesis, numbers of TC genesis linked to AEWs, and the ratio of TC genesis linked to AEWs to that total TC genesis during July to October 1984 to 2009.

| Region | Number of TC Genesis | Number of TC Genesis Linked to AEWs | Ratio of TC-AEWs to Total TCs (%) |
|---|---|---|---|
| East Atlantic | 89 | 69 | 77.25 |
| Mid-Atlantic | 88 | 56 | 63.64 |
| West Atlantic | 203 | 66 | 32.52 |
| Total | 380 | 191 | |

The results for all TC geneses and TC-linked AEWs are summarized in Table 1 by subdividing the main development region in the Atlantic into three: east Atlantic (east of 40° W), mid-Atlantic (60°–40° W), and west Atlantic (west of 60° W) c.f. [22]. In the table, TC genesis refers to location of the formation of named storms as identified in the HURDAT2. We note that the influence of AEWs on TC development is different for different regions over the Atlantic basin, e.g., [22].

Over the 26-year period, there were about 380 named TCs over the Atlantic basin, and 191 of the TC geneses were associated with AEWs. As shown in Table 1, TC genesis events increase westward-the majority of TC genesis occurs in the west Atlantic region (203, 53% of all TC geneses). About an equal number of TC geneses occurs in the East (89) and mid-Atlantic regions (88). Moreover, as reported in Hopsch et al. [22], TC genesis linked to AEWs increases eastward. About 77% and 64% of TC genesis is associated with AEWs over the east and mid-Atlantic basin, respectively. Eastern Atlantic TC development is more strongly influenced by AEWs compared with farther west c.f. [22]. Furthermore, about 33% of the genesis is associated with AEWs over the west Atlantic. Taken together, about

50% of Atlantic TCs are formed in association with AEWs, in agreement with past studies, e.g., [10]. We note that this is slightly less than a recent report by Russel et al. [15].

Consistent with past work, a significant portion of TC genesis occurred in association with propagating AEWs over the Atlantic. However, the impact of AEWs on TC development during the Atlantic hurricane season is also shown to be highly variable from year to year, e.g., [10,15]. Below, we examine the year-to-year variabilities of AEW and TC development.

The interannual variability of AEWs that crossed the West African coast (20°–15° W) between 1984–2009 is presented in Figure 3. As indicated above, coherent AEWs that crossed the West African coast during the study period were about 991, an average of about 38 waves per year. In the literature, the average number of AEWs generated over the continent during the boreal summer is about 50–60, e.g., [12,15,52]. For example, Pasch and Avila [16] noted that an average of about 59 AEWs formed over Africa during June-November 1967–1991. The difference with our result (average of 38 waves per year vs. 59) is expected, as our method considers only mature waves that crossed the western coast of West Africa and does not include waves that were initiated farther east and weakened before they reached the West African coast. The time period is also different. We considered July–October in contrast to the official hurricane season, June–November. On the other hand, Dieng et al. [24] reported 32 AEW troughs associated with convection that crossed the West African coast during July–September of 1990–2008. As shown in Figure 3a, of the total identified AEWs during July–October 1984–2009, about 20% of the matured AEWs were associated with TC genesis, while about 80% were not involved in TC genesis. The fraction of AEWs (~20%) that develop into TCs here are slightly higher than that reported by Avila et al. [12] and Agudilo et al. [17]. A brief analysis of Figure 3 shows that the annual number of TC-AEWs and the number of AEWs are weakly correlated. This is in agreement with past work such as Pasch and Avila [16], who suggested that the frequency of AEWs over Africa and TC genesis probability may not be correlated. As discussed above, most (~80%) of the AEWs are not linked to TC genesis.

Figure 3a also shows the marked year-to-year variability in coherent AEW crossings, ranging from 27 in 1991 to about 46 in 1985 and 2002. A brief inspection shows that AEWs that crossed coastal West Africa into the Atlantic were below average between 1991 and 1999. Except for 1994 and 1998, most of the below-average AEW crossings during the 1990s coincided with drier-than-average rainfall years over West Africa, e.g., [53]. As may be expected, AEW activities are weaker during the dry African years. Past work has also suggested that a weaker African rainfall season is related to weaker Atlantic TC activity, e.g., [54]. However, African wetter-than-average years do not always correlate with above-average AEWs. For example, 1994 and 2008 were wet years, e.g., [53], but AEW crossings were below average. The highest numbers of AEW-crossings (>1 standard deviation, σ) were observed in 1984–1985, 2001–2002, and 2004, while the lowest numbers of AEW-crossings (<1 σ) were observed in 1991, 1999, and 2007. Note the AEW crossings in 1995 and 2005 were 33 and 35, respectively, among the lowest, although these years were among active TC years.

The ratio (in percent) of named TCs that are linked to AEWs (black bars) to total AEWs that crossed the coast (red bars) are also shown in Figure 3b. The numbers of named TCs that are linked to AEWs are 15 in 1995 and 18 in 2005, both years being among the most active TC activity years, e.g., [55,56]. About 45% of the AEWs in 1995 and 51% of the AEWs in 2005 were instrumental in TC development (Figure 3b). While the number of AEW crossings in 2002 was among the highest, the AEW-TCs link was among the lowest (7%). The AEW-TC link was the weakest in 1997 (~3%, only one AEW triggered a named TC). Higher conversions of AEWs to TCs were also observed in 2008 (32%) and 1998 (29%) (see also [15]). We note that 1997 was one of the driest and 1998 was one of the wettest years over West Africa, e.g., [53].

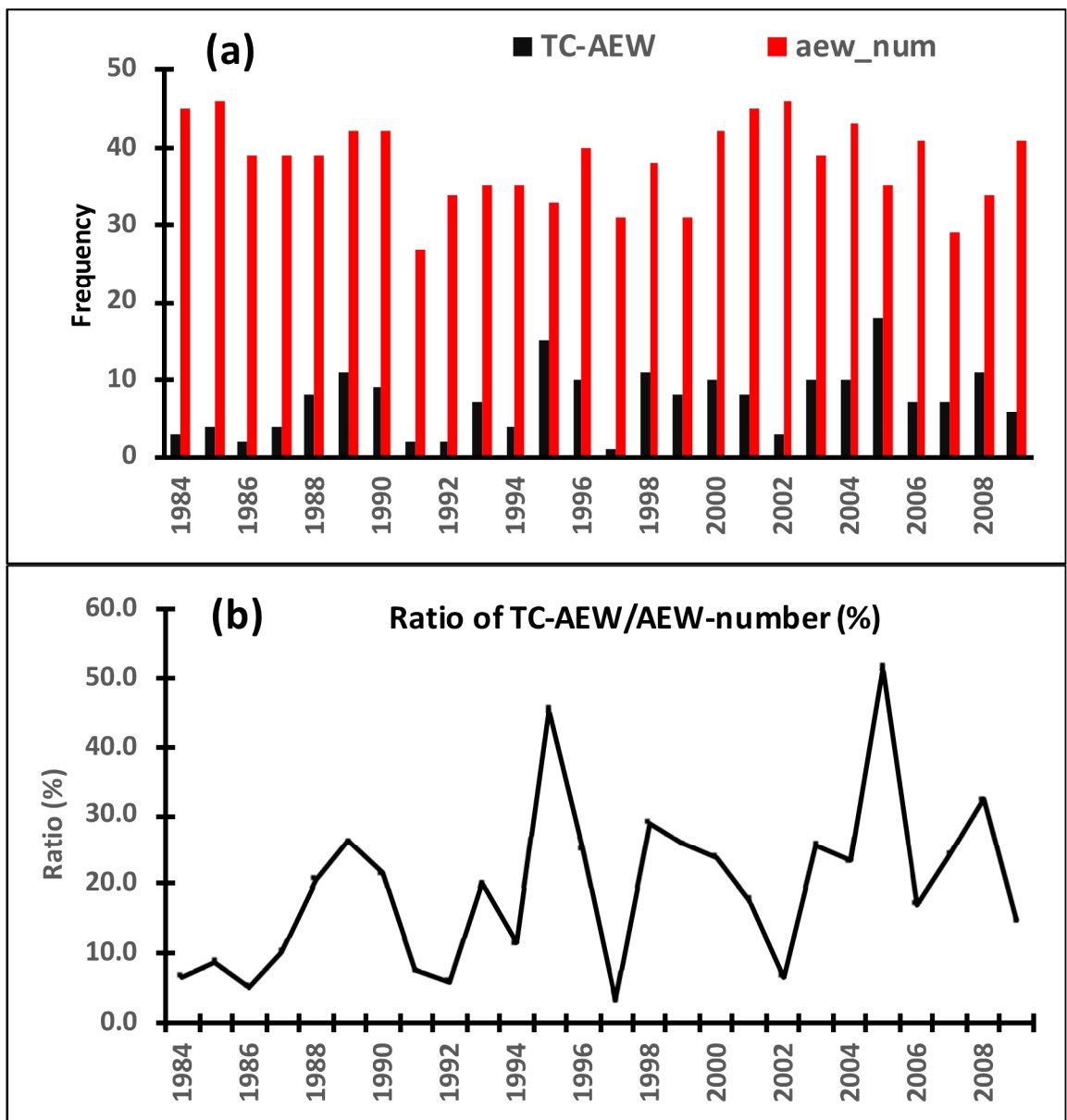

**Figure 3.** (**a**) The total number of AEWs (red bar) that crossed West Africa (20°–15° W) and named tropical cyclones spawned by AEWs (black bars). (**b**) The ratio (in percent) of named tropical cyclones spawned by AEWs and AEWs that crossed coastal West Africa.

The analysis above suggests that a higher frequency of AEW development over West Africa does not necessarily lead to a higher probability of TC development, e.g., [17,47]. Earlier study by Pasch and Avila [10] noted that the number of tropical waves does not coincide with the peak number of tropical storms over the Atlantic. Not all strong convectively coupled waves trigger tropical cyclones. This may be due to several other unfavorable large-scale environmental factors such as vertical wind shear and sea surface temperature (SST), e.g., [21]. Thorncroft and Hodges [47] reported that even if AEW activity were high, unfavorable deep shear and SST over the Atlantic could be detrimental for TC genesis (strong shear or colder SST). The role of the large-scale environment on TC development is well documented, e.g., [21]. Given a favorable large-scale environment, the potential explanation for a large fraction of coherent AEWs not triggering a large fraction of TC development may depend on the coupling between AEWs and different types of deep convection and the convective activity downstream of an AEW center. Leppert et al. [28]

noted that cloud cover associated with AEWs is an important factor for downstream TC development. They further suggested that the intensity of deep convection (intensity as measured by lightning flash rates) associated with the AEWs is not as critical compared with the wide area coverage of convection. We will investigate AEW-convection coupling using composites based on the ISCCP infrared weather state data and ERA Interim. As described above, the average states of the atmosphere (at different levels of the troposphere) are prepared using variables (e.g., humidity, wind, relative vorticity), separately, for developing and non-developing AEWs. This is accomplished below using dates and years of when coherent AEWs crossed the West African coast.

### 3.2. Relative Humidity and Wind Flow

In the literature, moisture availability to fueling deep convection associated with a propagating wave is described as critical (e.g., [22]). Recently, Russel and Aiyyer [57] suggested that moist convection is critical to AEW growth and propagation. Moist environment can be diagnosed using relative humidity, as an indicator of the moist environment, and the prevailing wind circulation associated with moisture. Therefore, we now briefly review the large-scale wind flow and relative humidity environments during developing and non-developing AEW events, e.g., [17,22]. Figure 4 presents composite anomalies (anomalies with reference to the July–October 1984–2009 mean; see also Figure 2c) of relative humidity (RH) and wind at 850 hPa and 700 hPa, separately, for developing and non-developing cases.

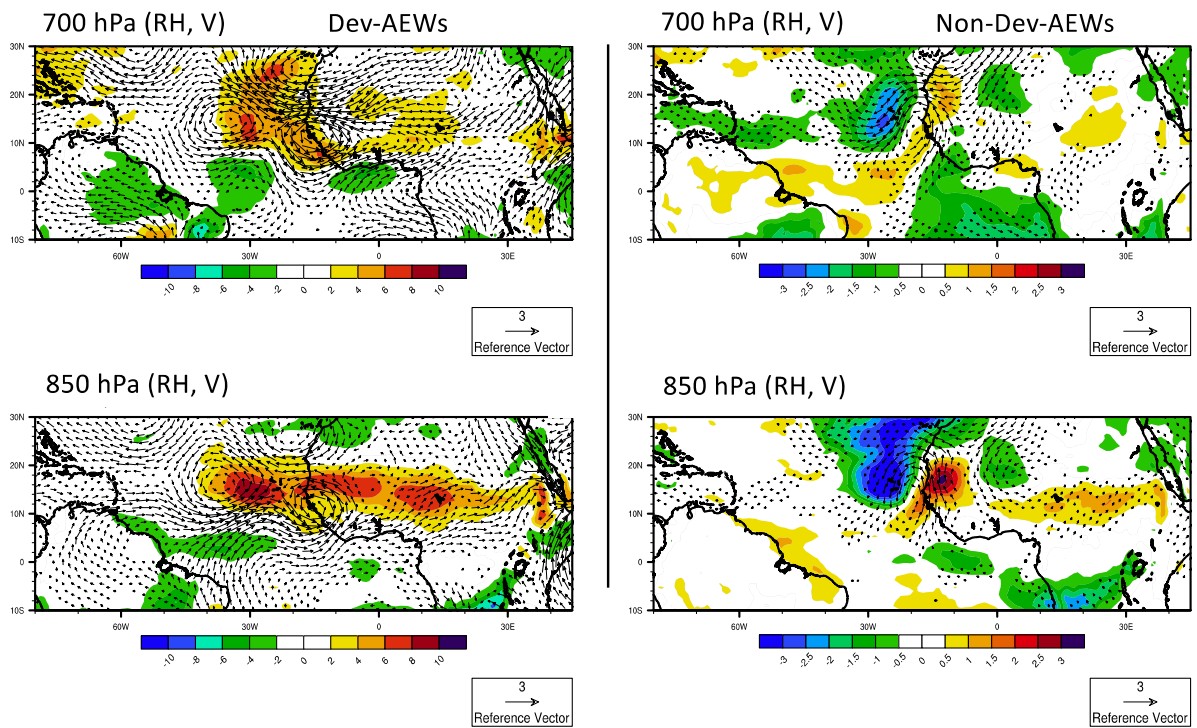

**Figure 4.** Relative humidity (shaded, in percent) and horizontal wind anomalies at 700 and 850 hPa for developing AEWs (**left** panel) and non-developing AEW (**right** panel) events. Anomalies are with respect to July–October 1984–2009 climatology. The magnitude of the reference vector is 3 ms$^{-1}$. Wind magnitudes of <0.2 ms$^{-1}$ are suppressed for clarity. Note the scale difference in relative humidity for developing and non-developing AEWs.

**Developing AEW composites (left panel)**: The composite for the developing AEW at 850 hPa shows an east-west elongated positive RH with peak values along 15° N. Peak RH values exceed 8% above average, indicating a moist environment conducive for deep convection ref. [23]. Above average RH anomalies are associated with anomalous cyclonic

circulation over coastal West Africa and the eastern Atlantic (Figure 4). At the 850 hPa level, the peak RH anomalies are to the north of the cyclonic center. The 700 hPa positive RH anomalies are seen over the coast and eastern Atlantic. While the 700 hPa RH anomalies are seen equatorward of the 850 hPa RH, the cyclonic circulation anomalies are nearly vertically aligned.

**Non-developing AEW composites (**Figure 4**, right panel):** Composites for non-developing AEWs at 850 hPa show positive RH anomalies along 15° W between 15°–20° N and between 10°–15° N east of the Greenwich meridian. Moreover, at 850 hPa, a negative RH anomaly is seen stretching north-south from around 30° N toward 10° N over the eastern Atlantic. The 850 hPa wind circulation associated with negative RH is observed over a wider area flowing equatorward from the Azores anticyclone. The flow is also associated with northeasterlies with characteristics of dry Saharan air, e.g., [58]. Note that the cyclonic circulation is also centered onshore over the eastern Atlantic, in clear contrast with the 850 hPa anomalous circulation for developing AEWs. The non-developing 700 hPa RH and anomalous circulation show a similar pattern as the 850 hPa (Figure 4).

Comparison of the environments for developing and non-developing AEWs in Figure 4 shows that the RH composite anomalies for non-developing AEWs are weak compared with developing AEWs, in agreement with past work, e.g., [22,23]. The 850 and 700 hPa wind circulations are also stronger during developing AEW events compared with non-developing AEW events. Tight circulation anomaly centers are seen over coastal West Africa during developing events, while a single circulation is seen just off the coast during non-developing events. The anomalous cyclonic circulation centers during the developing events are aligned more in the east-west direction, while the circulation anomalies in the non-developing cases are oriented in the southeast to northwest direction. Strong southward anomalies are seen off the continent deep toward the equator during non-developing events, while easterlies and southeasterly anomalies dominate the eastern Atlantic during developing cases. Northerlies and strong northwesterlies over the eastern Atlantic between 20°–40° W indicate the negative role of the strong anticyclonic circulation emanating from the Azores high pressure system. The RH composite anomalies for non-developing cases show a dipole-like pattern near 20° N at 850 hPa. Negative RH anomalies and strong northerlies ahead of the trough are indicative of an unfavorable environment for further wave development and deep convection, as suggested in previous studies, e.g., [23,57]. Over the east Atlantic the relative humidity anomaly is negative, which indicates dry air intrusion ahead of the trough. This could inhibit further growth of deep convection that would have contributed to TC genesis.

Above, we briefly surveyed the mean RH and wind environments for developing and non-developing events. Next, we present the evolution of developing and non-developing waves from a day before they exit the continent to two days after they exit coastal West Africa. Figure 5 presents composites of RH and 2–10 day filtered meridional wind at 700 hPa for developing (left panel) and non-developing (right panel) AEWs from day −1 to day +2. Recall that day 0 refers to dates when AEWs cross the west coast of Africa and day −1 refers to a day before AEWs arrive at the coast. The 2–10 day filtered meridional wind represents AEW structure, as in the past work, e.g., [59].

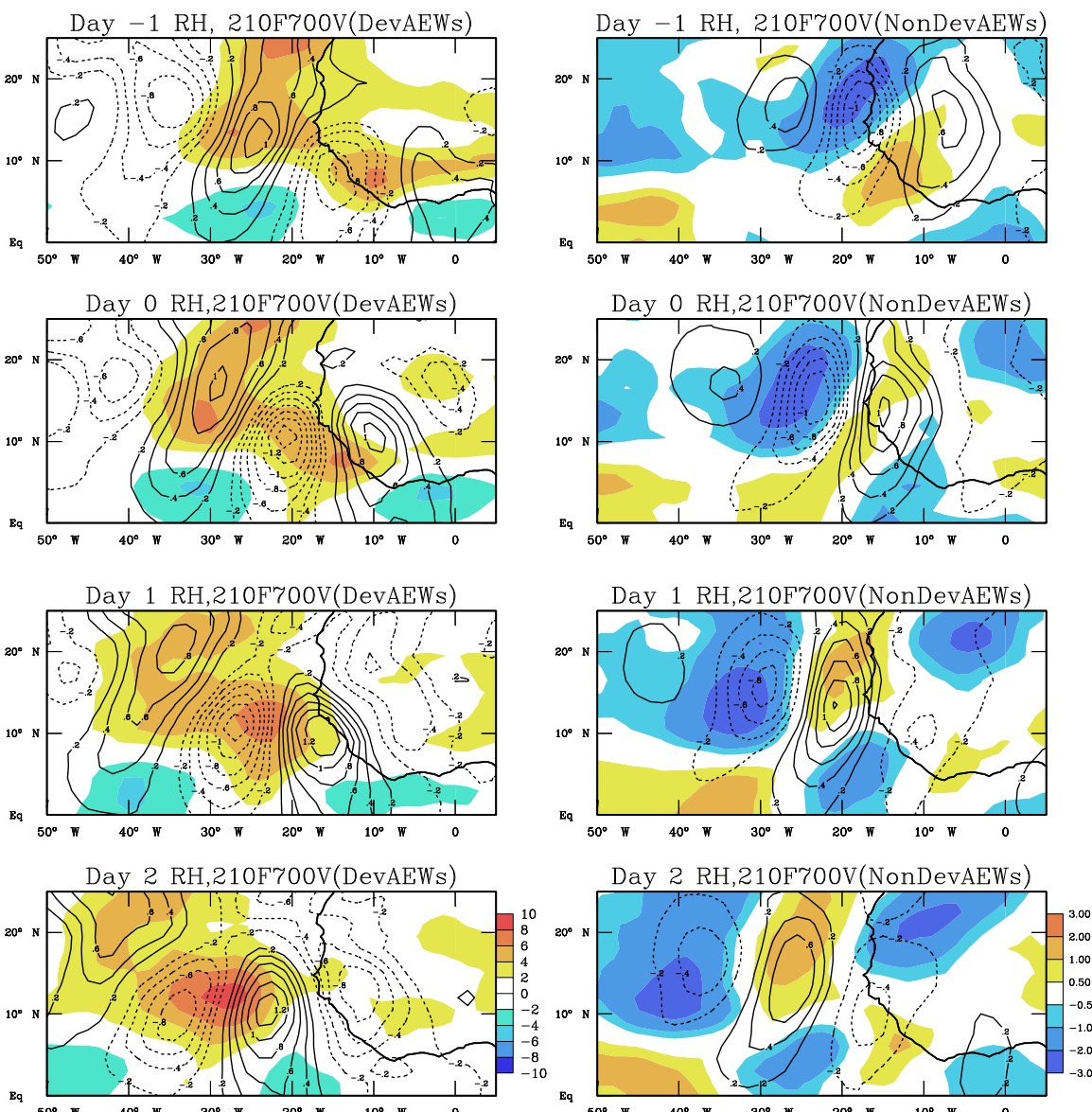

**Figure 5.** Composites of 700 hPa relative humidity anomaly (shaded, July–October 1984–2009 subtracted) and 2–10 day filtered meridional wind (contoured every 0.2 m/s, dashed for northerlies and solid lines for westerlies). Composites based on developing AEWs are shown on the (**left** panel). Composites based on non-developing AEWs are presented on the (**right** panel). Note the scale difference in the color bar between the left and right panels.

**Developing AEW composites (left panel):** At day −1, northerlies associated with positive RH anomalies dominate coastal West Africa. Positive RH anomalies associated with southerlies are seen over the eastern Atlantic north of 10° N. The trough axis at day −1 is located just to the east of 10° W around the Guinea highlands, indicating a conducive environment for further growth and development of AEWs and convective activity [6,22]. At day 0, the trough moved westward and was seen over coastal areas near 15° W. Maximum northerlies associated with positive RH anomalies (>4%) are seen along 20° W. Maximum positive RH anomalies are located slightly to the east of the trough axis over coastal areas and over the eastern Atlantic between 10°–20° N, in association with southerlies. Both positive RH anomalies and meridional winds appear to increase in their relative magnitude between day −1 and day 0. Furthermore, the downstream propagation of developing AEWs is accompanied by increasing positive RH anomalies (exceeding 6%) at day +1 and day +2. The waves propagate between 7–8° per day between day −1 and

day 2. In general, positive RH anomalies dominate the area downstream (westward) of the AEW trough, suggesting a favorable condition for tropical storm development. This result supports the conclusion from Figure 4 that strong southeasterly and easterly flow prevents dry northerlies associated with dry air from penetrating deep south.

**Non-developing AEW composites** (Figure 5**, right panel):** At day −1, a trough axis associated with positive RH (>1% above average) is observed near 10°–15° W and south of 15° N. The axis seems to move slowly westward and is situated just off land at day 0. At day 0, weak positive RH is seen near the trough axis and to the east of it, while negative RH anomalies dominate the region to the west of the trough axis. As the wave moves westward, the trough axis at day +2 is seen over the eastern Atlantic along 35° W, propagating 5–6° per day. A positive RH anomaly appears to be associated with southerly anomalies and is located to the east of the trough axis position. It is seen that the trough axis propagates westward into the RH environment that is lower than the climatology (negative RH anomalies) between days 1 and day 2. As seen in Figure 5, both meridional wind and RH anomalies during the non-developing AEWs are much weaker than those during the developing AEW cases (about 25% less than developing AEW events).

A significant feature of Figures 4 and 5 is that RH anomalies during the non-developing events are negative ahead (to the west) of the trough axis, while strongly positive during the developing AEW cases. The implication of this is that developing AEWs propagate through a middle troposphere that is moist and that is critical for sustaining convection at the area of large-scale deep convection (c.f. Figure 6). In contrast, negative RH anomalies indicate an unfavorable environment for sustaining deep convection during the non-developing events (e.g., [22,28,30]).

We also briefly inspected the deep vertical shear (200–850 hPa wind) and sea surface temperature (SST) over the Atlantic. Composites based on developing AEWs suggest that shear magnitudes are weaker (see Figure S2 in online Supplementary Material for vertical shear differences), and SSTs are warmer than the environment for the non-developing AEWs.

In Figures 4 and 5 above, we examined the lower and middle tropospheric wind flow and meridional winds, including relative humidity, to identify the large-scale impact on convection and the AEW structure. Next, we assess the interaction between different types of deep convection (using a direct measure in IR-WSs) and AEWs. Figure 6 presents, for both developing and non-developing AEWs, the interaction between different types of deep convection and the 700 hPa 2–10 day filtered meridional wind between days 0 and day 3. As discussed in Sections 2.1.3 and 2.2 above, well-organized deep convection is represented by IR-WS12. IR-WS12 is dominated by a mesoscale convective system (MCS) type of large-scale, well-organized deep convection and thick anvil clouds.

**Developing AEW cases** (Figure 6**, left panel):** At day 0, maximum deep convection over the coast is located within the trough axis, indicating a tight coupling between large-scale MCS convection and the AEW trough. As seen in the figure, a large area as far west as 30° W is covered by deep convection at day 0. The trough and maximum deep convection propagate westward and are seen west of 20° W at day 1 and around 30° W at day 2. At day 3, deep convection is centered along 35° W, but covering the area west of 40° W and near 30° W. Note that the peak northerlies and southerlies do not show weakening. Large area coverage of deep convection associated with a developing wave trough and northerlies seen in Figure 6 (left panel) is broadly in agreement with the findings of Leppert et al. [28], who suggested high coverage of deep convection in a developing wave trough and northerlies is critical for tropical storm development (see also [29,30]). Leppert et al. [28] reported that the high fractional coverage of convection is more important than the intensity of convection (as measured by lightning flash rates from a lightning imaging sensor). Nunez-Ocasio et al. [30] also concluded that the large area coverage of mesoscale convection has an important role for AEW mesoscale convection sustenance. However, it has to be noted that Leppert et al. [28] identified deep convection using top cold cloud brightness temperature ($T_B \leq 240$ K). Although a good proxy for convection, it is known that low $T_B$ cannot discriminate convective clouds associated with

large-scale mesoscale convective types (MCSs) from non-convective cirrus types in the upper troposphere, e.g., [31–33]. The cloud that covers a larger area that Leppert et al. [28] identified could be dominated by less well-organized deep convective types (IR-WS3) or a high percentage of cirrus clouds. It remains unclear how cirrus cloud types are identified and suppressed in the identification process. As we show here and below, the ISCCP WS system can identify different convective and non-convective states.

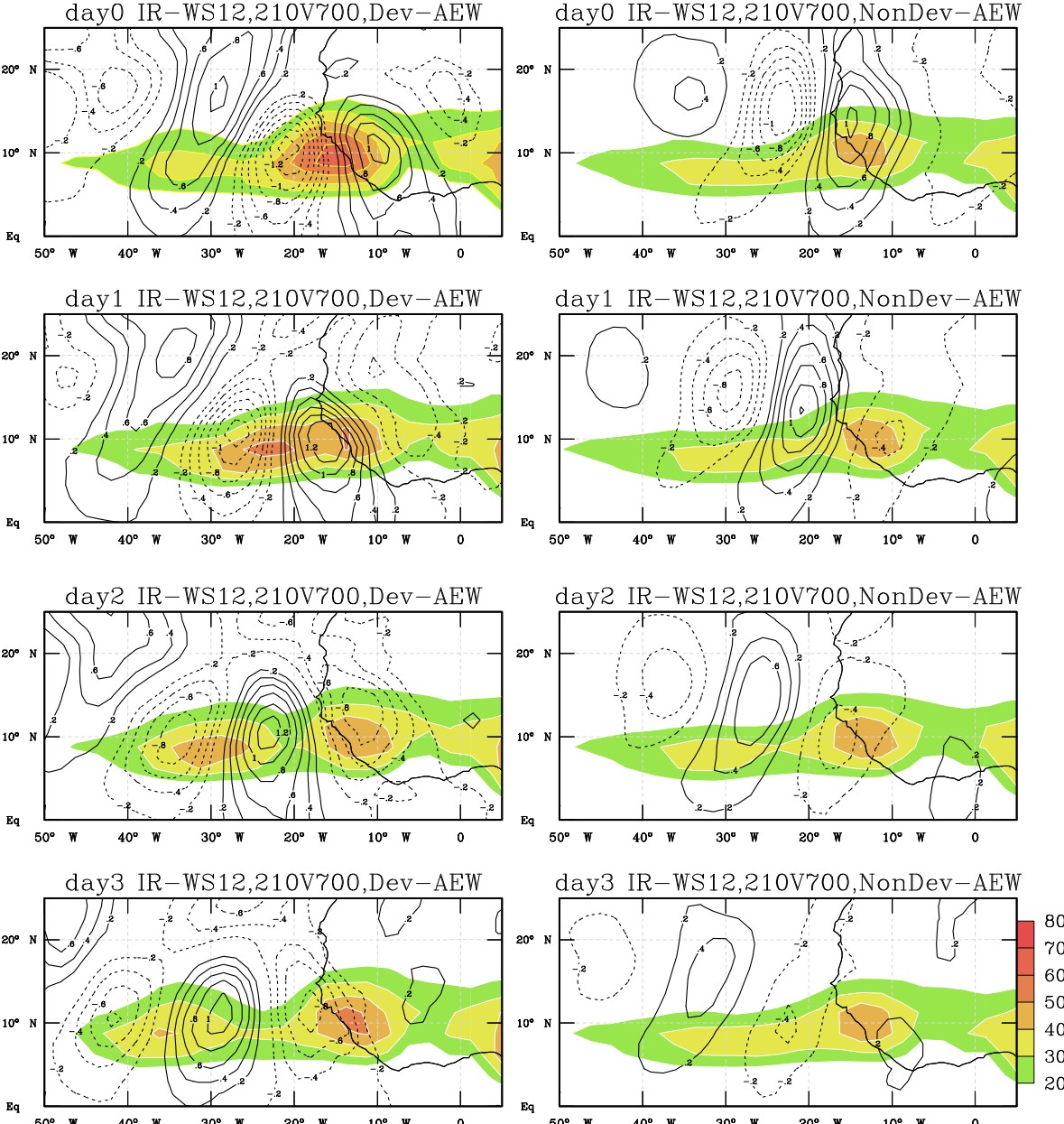

**Figure 6.** Composite of frequency of IR-WS12 (large-scale, well-organized deep convection) and 2–10 day filtered 700 hPa meridional wind for developing AEWs and for non-developing AEWs. IR-WS12 is shaded in color bars every 10%. Composites based on developing AEWs events are presented on the left, while composites based on non-developing AEWs are on the right side. Meridional winds are contoured every 0.2 m/s.

**Non-developing AEW cases** (Figure 6**, right panel):** The evolution of deep convection during the non-developing AEW events shows significant differences compared with composites based on developing AEWs. At day 0, the maximum deep convection is located

over land, while the wave trough is located offshore near 20° W. The wave trough propagates westward from day 0 to day 1 and 2, but deep convection progressively weakened, indicating the waning of the wave-deep convection coupling over time. In comparison, developing cases are distinguished by association with deep convection and tight coupling. Overall, Figure 6 shows that developing AEWs are tightly coupled with well-organized large-scale convection, while large scale well-organized deep convection is weak in a non-developing AEW environment. As discussed earlier, this coupling has an important effect on the role of AEWs triggering TCs in the eastern Atlantic.

Additionally, we examined the interaction between scattered, less well-organized deep convection (IR-WS3) and developing and non-developing AEW events, separately (Figure 7). Overall, high frequencies of scattered and less well-organized deep convective activities are located behind (to the east of) the developing AEW trough axis and about 2000 km (near 40° W) ahead (to the west) of the trough axis between day 0 and day 2. This feature is in contrast to the well-organized and large deep convection (IR-WS12), where the peak MCS type of convection is co-located with the trough axis and tightly coupled with the wave. Comparison between Figures 6 and 7 (left panels) also shows that the IR-WS3 types cover larger areas to the west of the trough axis (west of 40° W) than IR-WS12 types. For example, comparison of the locations of IR-WS12 and IR-WS3, with respect to the wave trough at day 1, shows that peak IR-WS3 convections are to the west of the peak IR-WS12 convections over the Atlantic (Figures 6 and 7 left panels). The prevalence of a high frequency of less well-organized deep convections far ahead of the wave trough is favorable for further wave growth and TC development downstream [24,25]. Our finding that IR-WS3 types cover a wider area than IR-WS12 is consistent with the conclusions of Leppert et al. [28], who suggested that a higher fraction of cloud coverage is more important than the intensity convection, although they did not explicitly discriminate different types of deep convection. We suggest the expansive cloud system that Leppert et al. [28] identified and located downstream from the AEW-MCS center is dominated by IR-WS3 type. Figure 7 also suggests that IR-WS3 is slightly weaker during the non-developing AEWs compared with IR-WS3 during developing waves.

As shown in the left columns of Figures 6 and 7, the developing AEW major trough axis located over coastal West Africa at day 0 propagated westwards and is seen just east of 40° W at day 3. This suggests that, on average, convectively coupled AEWs propagate 7–8° per day. Moreover, as shown in Table 1, the majority (77%) of the named tropical storms develop over the eastern Atlantic (east of 40° W) in association with AEWs. A comparison of this large percentage with the results in Figure 6 shows that TCs develop in the eastern Atlantic (east of 40° W) within 3 days of a developing AEW crossing coastal West Africa. Furthermore, to better understand the interaction between developing waves and different types of deep convection that influence TC genesis, we computed composites of developing AEWs and convective states using dates when TC genesis was first reported in the HURDAT2 database (c.f. Table 1), and the results are summarized below.

Figure 8 shows the composite frequency of IR-WS12 and 2–10 day filtered 700 hPa meridional wind ($v$) based on reported dates of the first TC genesis over the region east of 40° W. About 55 AEW-linked genesis cases were identified. Therefore, day 0 corresponds to composites of IR-WS and $v$ at the time of AEW-linked TC genesis. At day −1 (a day before TC genesis), the maximum frequency of IR-WS12 (large-scale, well-organized deep convection) is located within the trough near 30° W, stretching eastwards to coastal West Africa. Both well-organized deep convection and the AEW trough coherently move westwards and are located west of 30° W at day 0. The maximum IR-WS12 occupies a large area around the trough and slightly behind the trough between 40° W and 30° W (Figure 8, middle panel). We note that the frequencies of IR-WS12 at day −1 are slightly higher than at day 0, suggesting a favorable AEW-convection coupling for TC genesis a day 0. At day 1 (a day after TC genesis), while the developing AEW moves westwards, maximum deep convection still dominates the area near and to the east of the trough and in-phase with southerly anomalies, consistent with early work, e.g., [7,8,18].

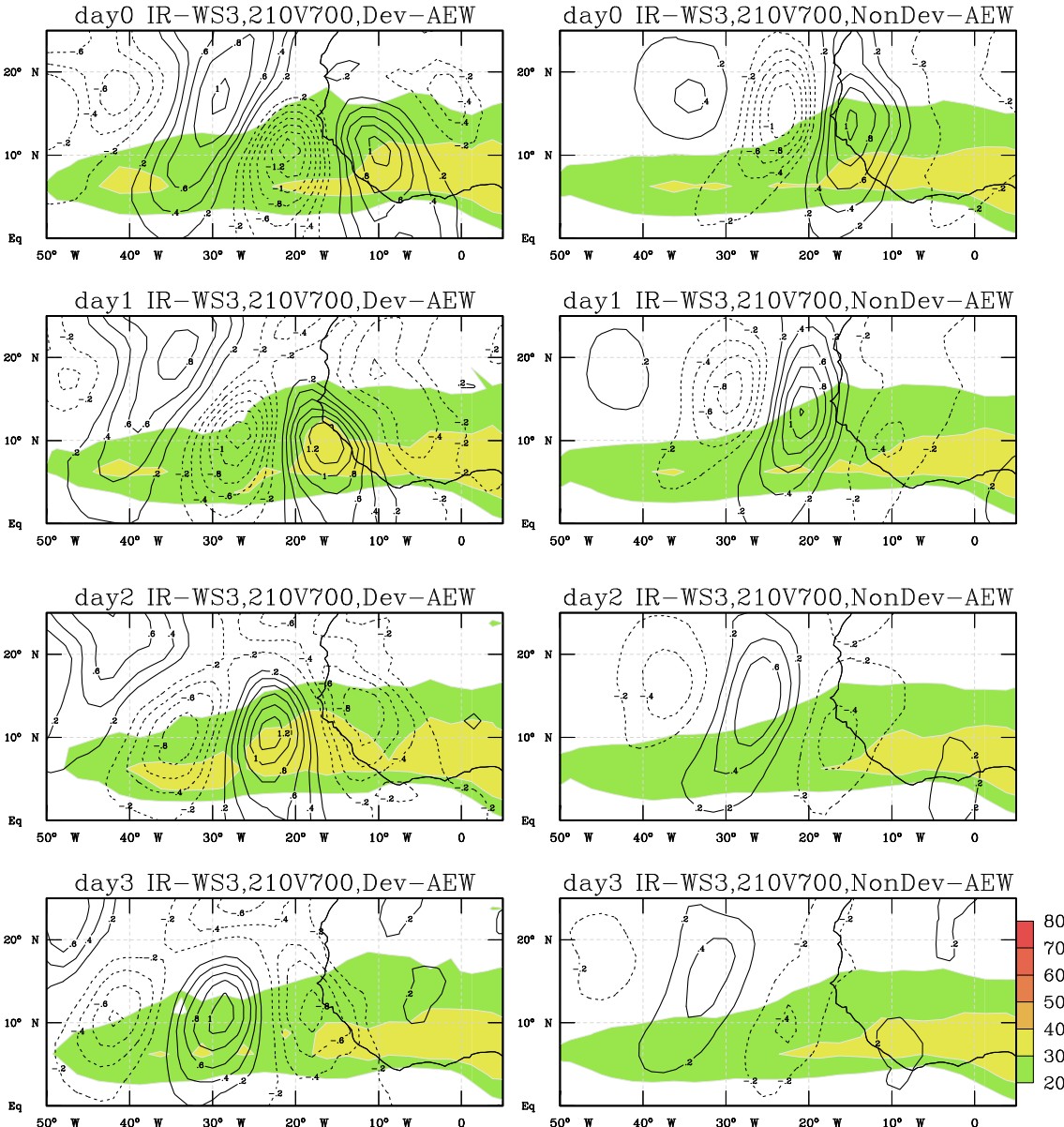

**Figure 7.** Composite of frequency of IR-WS3 (smaller scale less well-organized deep convection) and 2–10 day filtered 700 hPa meridional wind for developing AEWs and for non-developing AEWs. IR-WS3 is shaded in color bars every 10%. Composites based on developing AEW events are presented on the left, while composites based on the non-developing AEWs are on the right side. Meridional winds are contoured every 0.2 m/s.

The interactions between IR-WS3 and AEWs during the TC genesis events were also examined (Figure 9). Scattered convection (IR-WS3) is seen over land around the Guinea highlands and adjoining the eastern Atlantic before TC genesis and a day after genesis [6]. A day before TC genesis and on the day of genesis, higher frequency and coverage of scattered and isolated deep convection are seen east and south of the trough axis, in agreement with the findings of Nunez-Ocasio et al. [29,30]. The maximum frequency of IR-WS3 is located about 500 km to the south and west of the maximum IR-WS12, suggesting a conducive environment for westward propagation and growth of wave disturbances and tropical storms. Past work suggested that deep convection covering a larger area ahead of the wave axis is conducive for further wave and convective growth and maintenance. This suggests that a large area of cloud coverage would increase the likelihood of TC growth in association with wave disturbances, e.g., [24,28,30,60].

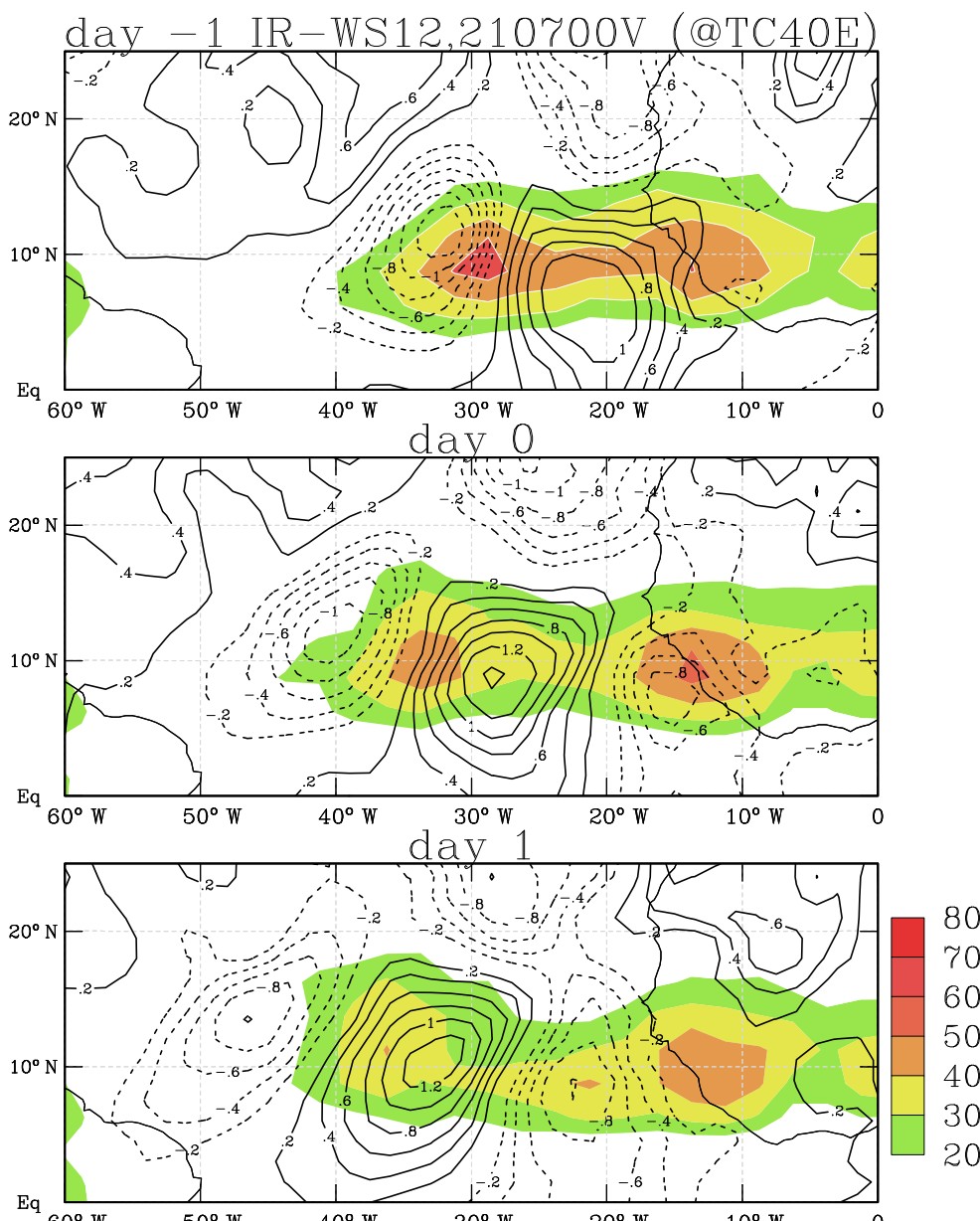

**Figure 8.** Composites of IR-WS12 (shaded every 10% from starting from 20%) and 2–10 day filtered 700 hPa meridional wind (contoured every 0.2 m/s). Composites are constructed based on dates of tropical cyclone genesis over the region 40°–20° W.

Studies have shown that AEWs coupled with deep convection have a vertical structure that extends up to the tropopause, e.g., [61,62], and are likely to effect TC development. For completeness, we briefly investigated the vertical structure (surface-200 hPa) of developing and non-developing AEWs using relative humidity and meridional wind fields. Results (not shown) confirm that a positive relative humidity anomaly in the zonal-vertical cross section is evident in a wide region between 30° W and 30° E for developing waves, while only a narrow area centered at 15° W is seen for the non-developing waves.

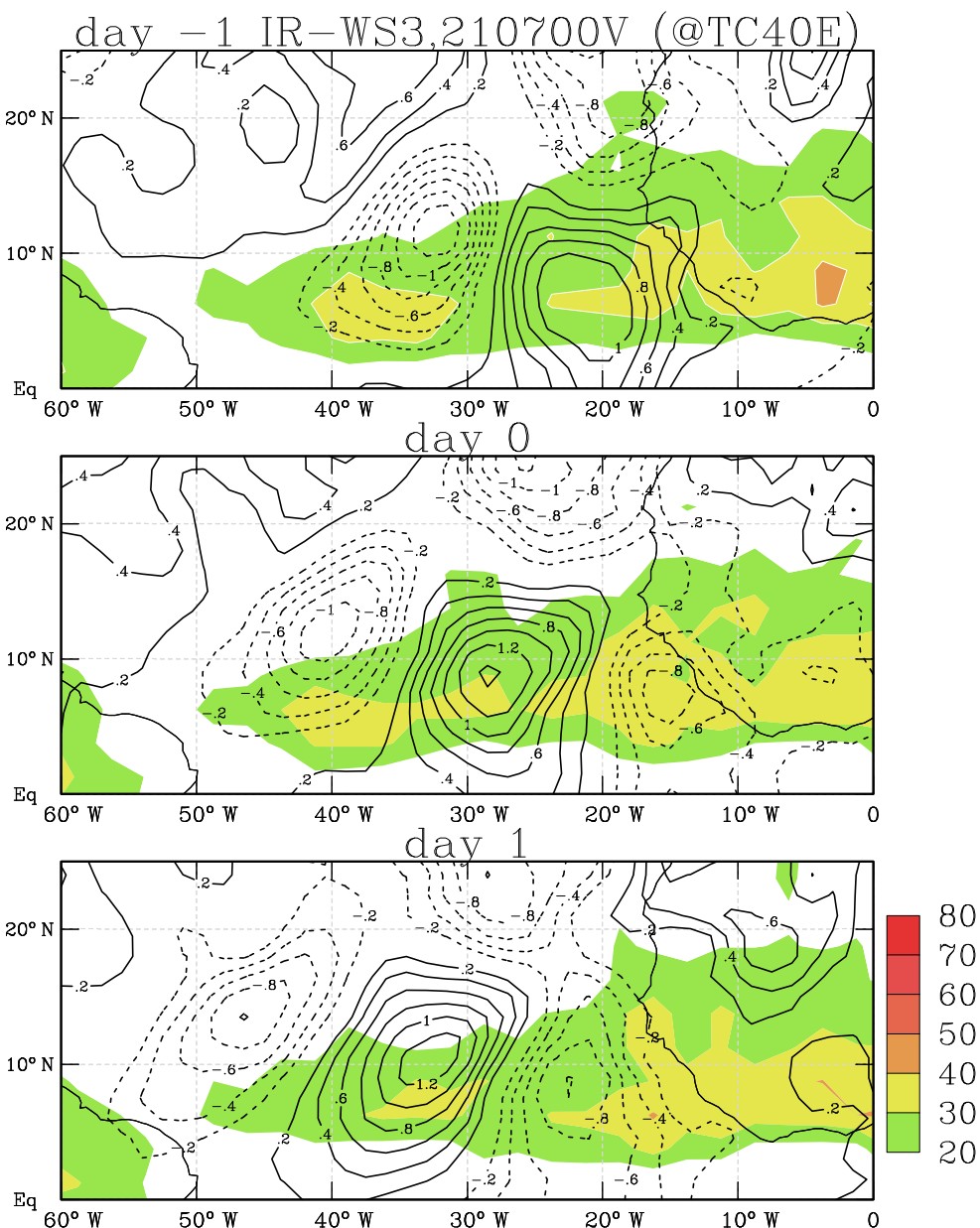

**Figure 9.** Composites of IR-WS3 (shaded every 10% from starting from 20%) and 2–10 day filtered 700 hPa meridional wind (contoured every 0.2 m/s). Composites are constructed based on dates of tropical cyclone genesis over the region 40°–20° W.

## 4. Summary and Remarks

Past studies suggested that only a small fraction of AEWs that form over Africa serve as precursors to TC genesis in the Atlantic. Given this, the main goal of this study was to investigate why or how a small fraction of AEWs do spawn tropical cyclones while a high proportion of them do not. We also briefly revisited the large-scale environment that AEWs propagate through between the east Atlantic and west coast of Africa. We achieved our objective using National Hurricane Center best track data, ISCCP weather states, and ECMWF Interim reanalysis (ERA-I) products. Mature AEWs were identified using an objective tracking of vorticity centers following the algorithm developed by Hodges (50–51) and Thorncroft and Hodges [47], based on the ERA-I dataset. A total of 991 AEWs were identified during the study period (July-October of 1984–2009), out of which ~20% of the AEWs were associated with tropical cyclogenesis (developing waves) and the remaining 80% of the AEWs were not associated with TC genesis (non-developing waves). The total

number of AEWs per year revealed marked interannual variability. Overall, the number of AEWs in dry African years was smaller than in wet rainfall years. It is also noted in some years (e.g., 1995, 2005) that the number of waves propagated from West Africa was slightly lower than average, but the rate of conversion to TC genesis was high. Of the total TCs generated in the main development region, about 55% generated were associated with AEWs, in agreement with previous findings, e.g., [10,12,55]. TC genesis over the east and mid-Atlantic basin is strongly related to AEWs (77% and 64%, respectively). The fact that only a small fraction of AEWs serve as precursors of named storms suggests that the larger scale environment that the waves propagate through plays an important role for TC genesis. Additionally, as we describe below, for an AEW to develop into a named storm, the coupling of AEWs with large-scale, well-organized deep convection and an environment that is dominated by deep convection ahead of the trough are important conditions. On average, non-developing waves lack connection to large-scale, well-organized convection.

Composites of relative humidity (RH) and wind anomalies based on developing AEWs showed a dominance of high positive RH over a larger area around the AEW axis and to the west of it. The developing AEWs were also associated with strong circulation that is favorable for TC genesis over the Atlantic basin. Composites based on non-developing AEWs showed negative RH anomalies ahead of the wave trough that would limit the likelihood of TC genesis. Moreover, a negative relative humidity anomaly in the middle and lower troposphere is an indicator of a lack of moisture for wave growth and development while propagating downstream in the eastern Atlantic basin.

Composites based on the dates of AEWs that departed the western coast of Africa revealed differences in the characteristics of the interaction between different types of deep convection and developing and non-developing AEWs, which has implications for TC development. In this study, different types of deep moist convection were represented using an infrared weather state (IR-WS), which provides a unique platform to distinguish different types of moist convection and their coupling with atmospheric waves. Composites of large and well-organized deep convection, including thick anvil clouds (IR-WS12) and mid-tropospheric meridional wind (as an AEW measure), revealed that developing AEWs are strongly coupled with deep convection in the lower and middle troposphere. On the other hand, scattered and less well-organized deep convections (IR-WS3) are prevalent and cover a larger area downstream of the AEW axis over the main development region in the eastern Atlantic ahead of the AEW axis. These results suggested that while the association of developing AEWs with the large-scale deep convection is critical for the strengthening of AEWs, the coverage of a wide area ahead (to the west) of the wave by IR-WS3 types is important for wave growth and development downstream, e.g., [24,25,28]. We suggest that distinguishing between different deep convective types is an important input for waves and TC genesis outlook. This is a significant aspect of this study.

Additionally, to shed light on the prevailing environmental conditions, we investigated the interaction between two different types of deep convection and developing AEWs based on the first appearance of a named storm genesis. In other words, we extended our investigation by constructing composites of IR-WS12, IR-WS3, and AEWs using the first dates of TC genesis in the eastern Atlantic, east of 40° W. The results showed that large-scale and well-organized deep convection coupled with AEWs dominated the eastern Atlantic and West Africa during the TC genesis events and one day before TC genesis. The dominance of deep convection over coastal West Africa and the eastern Atlantic a day early supports the growth and strengthening of AEWs and favors TC genesis on day 0. The comparison of IR-WS12 convection between day −1 and day 0 showed that deep convective activity at day −1 was stronger than deep convection at day 0 (the time of TC genesis). At the same time, the area downstream of the peak AEW activity and maximum deep convection was dominated by IR-WS3 (scattered and less well-organized deep convection), consistent with the composites based on departing AEWs from coastal West Africa. The IR-WS3 coverage extended up to about 500 km westwards from the AEW–deep convection center. This result qualifies the past conclusion that stronger AEWs and expansive cloud

coverages downstream are crucial for wave development and growth and, ultimately, TC genesis in the eastern Atlantic, e.g., [24,28]. Note that we arrived at this conclusion based on two different methods of composite: using the dates of AEW exit from coastal West Africa and the first days of named storms as reported in HURDAT2.

We suggest that identifying and monitoring the types of deep convection that are interacting with waves, including locations with respect to a wave axis (around the wave axis and ahead of it), could provide insight into the TC genesis forecast effort.

**Supplementary Materials:** The following are available online at https://www.mdpi.com/article/10.3390/atmos13010005/s1. Figure S1: The ISCCP Cloud Classification: The nine basic cloud classification describe low-, middle and high-level clouds. This is similar to the World Meteorological Organization's (WMO) cloud classification. The WMO classify Cumulonimbus as the tenth cloud genera. More details can be found at https://public.wmo.int/en/media/news/international-cloud-atlas (accessed on 13 December 2021), Figure S2: The vertical deep shear (200–8500 hPa) difference between developing and non-developing AEW events. Negative implies the wind shear magnitude during developing AEWs is weaker than non-developing AEW events. Day 0 indicates the time of AEWs as they exit the coast of West Africa and enter into the Atlantic. Day 1 represents the shear difference after waves exit West Africa and are located in the Atlantic. Contours are every 0.5 ms$^{-1}$.

**Author Contributions:** B.D.E. contributed to writing and computation, including visualization. A.M. provided conceptualization, funding acquisition, and writing. All authors have read and agreed to the published version of the manuscript.

**Funding:** The initial stage of this research was funded by the National Science Foundation under grant number AGS-1461911.

**Institutional Review Board Statement:** Not applicable.

**Informed Consent Statement:** Not applicable.

**Data Availability Statement:** Data processed and used for this research can be made available by request to authors.

**Acknowledgments:** We thank Kevin Hodges, the National Centre for Atmospheric Research, University of Reading, for providing his tracking code and for his instructions. The initial stage of this research was supported by the National Science Foundation under grant number AGS-1461911. We thank two anonymous reviewers for their insightful comments and suggestions, which greatly improved this paper.

**Conflicts of Interest:** The authors declare no conflict of interest.

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
