# Peer review of "The Interaction between African Easterly Waves and Different Types of Deep Convection and Its Influence on Atlantic Tropical Cyclones"

_atmosphere, doi:10.3390/atmos13010005_

Round 1
Reviewer 1 Report
Please find review in Word document

Author Response
We thank our reviewer for a comprehensive and detailed comments which improved our manuscript significantly.

Reviewer 2 Report
The article ‘The interaction between African easterly waves and different types of deep convection and its influence on Atlantic tropical cyclones’ qualitatively analyzes the correlation between African Easterly Wave (AEWs) to developing and non-developing Atlantic tropical cyclone cases. To strengthen their argument, the authors used additional factors such as Relative Humidity and wind flow associated with the events. Th study is extensive and adds insights to the AEW and TC correlation theory. I feel the article worth publication with consideration of the following minor queries and corrections.
- Please capitalize, 'African Easterly Waves (AEWs)' in abstract.
- The objectives are not clear in the introduction
- Font size in the figure labels is not identical with the font of the figure or the text size
- The analysis is based on RH anomalies however, did not quantify it. It would be worth mentioning if the anomaly is spatio-temporal, if so, what are the time and space domains.
- It would be worth investigating the time lag between AEW and TC genesis.
Author Response
We thank our reviewer for their time and ideas which improved our manuscript.

Round 2
Reviewer 1 Report
Accept.
Reviewer 2 Report
The manuscript is much improved with recent citations. The review response are convincing. After minor correction this manuscript is recommended for publication. I am still not convinced why some of the titles and legends for colorbars have much bigger font. Please use uniform format in the text, e.g., in Line 289 a space is there before % and no space in many other places.
Line 315: The total TC variance is also weakly - What does this sentence mean? The sentence itself is incomplete.
Replace African Easterly Wave (AEWs) with African Easterly Wave (AEW)s.
Author Response
We thank our Reviewers for their insightful comments, which improved the manuscript. Below is a response for the second review (Reviewer comments in red and italics).
Reviewer 2:
The manuscript is much improved with recent citations. The review response are convincing. After minor correction this manuscript is recommended for publication.
I am still not convinced why some of the titles and legends for colorbars have much bigger font. Please use uniform format in the text, e.g., in Line 289 a space is there before % and no space in many other places.
Editorial is done (see track changes)!
Line 315: The total TC variance is also weakly - What does this sentence mean? The sentence itself is incomplete.
We are sorry about this. It looks like we uploaded file before saving.
Replace African Easterly Wave (AEWs) with African Easterly Wave (AEW)s.
Done.